# Efficient combinatorial optimization by quantum-inspired parallel annealing in analogue memristor crossbar

Mingrui Jiang [1], Keyi Shan[1], Chengping He[1] & Can Li [1] ✉

Combinatorial optimization problems are prevalent in various fields, but obtaining exact solutions remains challenging due to the combinatorial explosion with increasing problem size. Special-purpose hardware such as Ising machines, particularly memristor-based analog Ising machines, have emerged as promising solutions. However, existing simulate-annealing-based implementations have not fully exploited the inherent parallelism and analog storage/processing features of memristor crossbar arrays. This work proposes a quantum-inspired parallel annealing method that enables full parallelism and improves solution quality, resulting in significant speed and energy improvement when implemented in analog memristor crossbars. We experimentally solved tasks, including unweighted and weighted Max-Cut and traveling salesman problem, using our integrated memristor chip. The quantum-inspired parallel annealing method implemented in memristor-based hardware has demonstrated significant improvements in time- and energy-efficiency compared to previously reported simulated annealing and Ising machine implemented on other technologies. This is because our approach effectively exploits the natural parallelism, analog conductance states, and all-to-all connection provided by memristor technology, promising its potential for solving complex optimization problems with greater efficiency.

Combinatorial optimization problems (COPs) are ubiquitous in social life and industry, with diverse applications spanning computer science, engineering, chemistry, logistics, economics, and beyond[1–3]. Unfortunately, exact solutions are notoriously challenging to obtain due to the combinatorial explosion that occurs as problem size increases. Consequently, special-purpose hardware such as Ising machines are increasingly sought after to efficiently solve COPs by mapping them to the Ising model, a statistical model that describes a physical system comprising spins that interact with one another, which tends to evolve into the lowest system energy[4,5]. Although classical computers can emulate the process, their efficiency is significantly limited due to their digital and serial nature and the property of separated memory and processing units. To overcome these limitations, researchers have explored various analog technologies

including superconducting qubits[6,7], coherent lights[8–10], CMOS oscillators[11,12], nano oscillators[13–15], and memristors-based analog Ising machines[16–18]. Among these, memristor-based machines offer particular promise with their speed and energy efficiency, natural all-to-all connections, and compatibility with electronic computing ecosystems.

Nonetheless, existing memristor-based demonstrations have not fully exploited the massive parallelism and analog storage/processing features of memristor crossbar arrays. Previous works mainly relied on simulated annealing and its variants to obtain the optimal solution and thus were limited to the serial updating nature of simulated annealing[16–23]. Consequently, in these implementations, the memristor crossbar performed only one vector-vector dot-product operation at a time, rather than a vector-matrix multiplication, resulting in a huge

---

[1]Department of Electrical and Electronic Engineering, The University of Hong Kong, Hong Kong SAR, China. ✉e-mail: canl@hku.hk

waste of the natural parallelism provided by the memristor crossbar array. Adiabatic annealing as another annealing method with solid theoretical background which has successfully implemented in quantum systems, recently has demonstrated effectiveness with classical memristor-based platforms[24], shedding light on possible parallel updates. However, in this pioneer work, simulated annealing is still required in the solving process in addition to adiabatic annealing and thus does not get rid of its serial updating limitation[24]. Moreover, most previous works only demonstrated binarized memristor conductance, instead of analog values to represent arbitrary coupling strength in representing the problem, limiting the complexity of the problem that it can solve[25].

In this work, we demonstrated quantum-inspired parallel annealing (QPA) implemented in analog memristor crossbar can improve the solution quality and enable full parallelism, leading to significant speed and energy improvements. To minimize the system energy while compatible with the QPA, we developed a constrained gradient-descent method that was inspired from the training process of binary neural networks (BiNNs). Various tasks including 64-node unweighted, weighted Max-Cut and nine-city traveling salesman problem (TSP) were experimentally solved with the novel QPA scheme on our integrated memristor chip. By fully leveraging the advantages brought by the new technology, including natural parallelism, analog conductance states, and all-to-all connection, our memristor-based QPA demonstrated significant performance improvement. Our approach yielded a 2.3× speed benefits compared to the fastest reported results from memristor-based Hopfield networks (mem-HNN), 3.1× increase in energy efficiency over the most energy-efficient phase-transition nano-oscillators (PTNO) based continuous-time dynamic system reported in the literature, and orders of magnitude advantages than other technologies, including oscillator, digital, coherent light, and quantum systems.

## Results

### Quantum-inspired parallel annealing (QPA)

COPs can be reformulated as Ising models. To solve the problem is to find the ground state of a certain energy function known as Ising Hamiltonian, expressed as

$$H_{\text{Ising}} = -\sum_{i<j} J_{ij}\sigma_i\sigma_j - \sum_i h_i\sigma_i = -\frac{1}{2}\boldsymbol{\sigma}^T\boldsymbol{J}\boldsymbol{\sigma} - \boldsymbol{h}^T\boldsymbol{\sigma} \quad (1)$$

where $\boldsymbol{\sigma} = \{\sigma_1, \sigma_2, \ldots, \sigma_i, \ldots, \sigma_N\}$ is the spin configuration vector that encodes the problem's solution, with each component $\sigma_i$ being a binary value {−1, 1} representing the two states of spin-up and spin-down. $\boldsymbol{J}$ is a square and symmetric coupling matrix of size $N \times N$, with each element representing the ferromagnetic or antiferromagnetic connections between two spins. $\boldsymbol{h}$ is a local-field term introduced for generality.

In physical quantum systems, the annealing process to solve such a model is accomplished by adiabatic evolution[26–28]. It starts with a simple Hamiltonian $H_{\text{initial}}$, of which the ground state can be easily found (e.g., a transverse field Hamiltonian $H_{\text{initial}} = -\sum_i \sigma_i^x$)[26,27,29] and gradually shifts to the Ising Hamiltonian $H_{\text{Ising}} = -\sum_{i<j} J_{ij}\sigma_i^z\sigma_j^z$, expressed as

$$H_{\text{system}}(t) = A(t)H_{\text{initial}} + B(t)H_{\text{Ising}} \quad (2)$$

With A(t) gradually changes from 1 to 0 and B(t) gradually changes from 0 to 1. Ideally, a physical system can always retain the minimum-energy state and thus the system can eventually evolve to the ground state of the Ising Hamiltonian and solve the corresponding COP.

However, in this quantum version of adiabatic annealing, the spin is represented by a Pauli matrix rather than a discrete value, which is not suitable for our current classical analog memristor crossbar to

emulate. Therefore, we adjusted it for easier implementation in our analog memristor crossbar and proposed a new classical version of adiabatic annealing, in which the spin $\sigma_i$ is represented by a discrete value, either 1 or −1. In order to conduct annealing, an analog variable $x_i$ is introduced to represent the intermediate spin states and can be deemed as a classical "superposition" of the spin. The real spin configuration of $\sigma_i$ is taken as the sign of $x_i$. Similarly with the quantum version, we implemented a time-dependent system Hamiltonian:

$$H_{\text{system}}(t) = \lambda(t)H_{\text{initial}} + H_{\text{Ising}} \quad (3)$$

where, $\lambda(t)$ is a time-dependent coefficient, which starts with a sufficiently large value and gradually decreases to 0 during the solving process. $H_{\text{initial}}$ is the initial Hamiltonian and can be arbitrary function the ground state (global minimum) of which can be easily found. In this work, we chose $H_{\text{initial}} = \frac{1}{2}\sum_i x_i^2$, one of the simplest convex functions (different choice of $H_{\text{initial}}$ and their comparisons can be found in Fig. S1). In the solving process, the system is first dominated by $H_{\text{initial}}$ and will gradually shift to $H_{\text{Ising}}$, thus leading to the ground state of $H_{\text{Ising}}$. However, unlike real physical systems that naturally tend to evolve towards the lowest energy state during adiabatic evolution, our adiabatic Hamiltonian shift requires the development of an algorithm to enable manual updating of spins and thus lower the energy.

To address this challenge, we applied gradient descent, a popular optimization technique used in artificial neural network training[30], to help the system dynamically evolve into lower system Hamiltonian during adiabatic shift. However, vanilla gradient descent cannot be applied in this case due to the discrete nature of the spin configuration. Each update can only flip the sign of the spin configuration, leading to divergence. To overcome this limitation and inspired by the training of binary neural networks (BiNN), we turned to the straight through estimator (STE) algorithm, a modified version of gradient descent that shows great success in the field of neural networks[31–33]. The key idea behind STE is to introduce a full-precision "latent" weight as a proxy for the binary weight, which is binarized in the forward and backward path to calculate the gradient value. This gradient value is then directly used as an estimator to update the full-precision "latent" weight. Interestingly, the analog intermediate spin variable $x_i$ that we introduced earlier to represent the classical "superposition" of the spin states shares similar properties with the "latent" weight and can thus serve as a proxy for the spin.

To implement the STE algorithm in our Ising machine, we first projected the analog spin variable $\boldsymbol{x}$ onto the binary domain using a sign function to obtain the real spin configuration $\boldsymbol{\sigma}$. The real spin configuration was then used to calculate the gradient of the system Hamiltonian. Specifically, the gradient is given by

$$\text{gradient} = \nabla H_{\text{system}} = \nabla H_{\text{Ising}} + \lambda(t)\nabla H_{\text{Initial}} = -\boldsymbol{J}\boldsymbol{\sigma} - \boldsymbol{h} + \lambda(t)\boldsymbol{x} \quad (4)$$

We then used this gradient to update the analog spin proxy $\boldsymbol{x}$ as

$$\boldsymbol{x}(t+1) = \boldsymbol{x}(t) - \eta * \text{gradient} \quad (5)$$

where $\eta$ is the step size. To improve the performance of the algorithm, we have implemented two commonly used techniques from stochastic gradient descent in neural network training: clipping and momentum, which are widely used in stochastic gradient descent in neural network training. More techniques from modern neural networks can potentially be applied to further improve performance in the future. For more details about the algorithm that we have implemented, please refer to the Methods section and more discussion and comparison of the updating techniques can be seen in Fig. S2. Notably, the gradient-based updating was simultaneously applied with the adiabatic Hamiltonian shift to dynamically find the ground state of the current system Hamiltonian and eventually evolve to the ground state of the

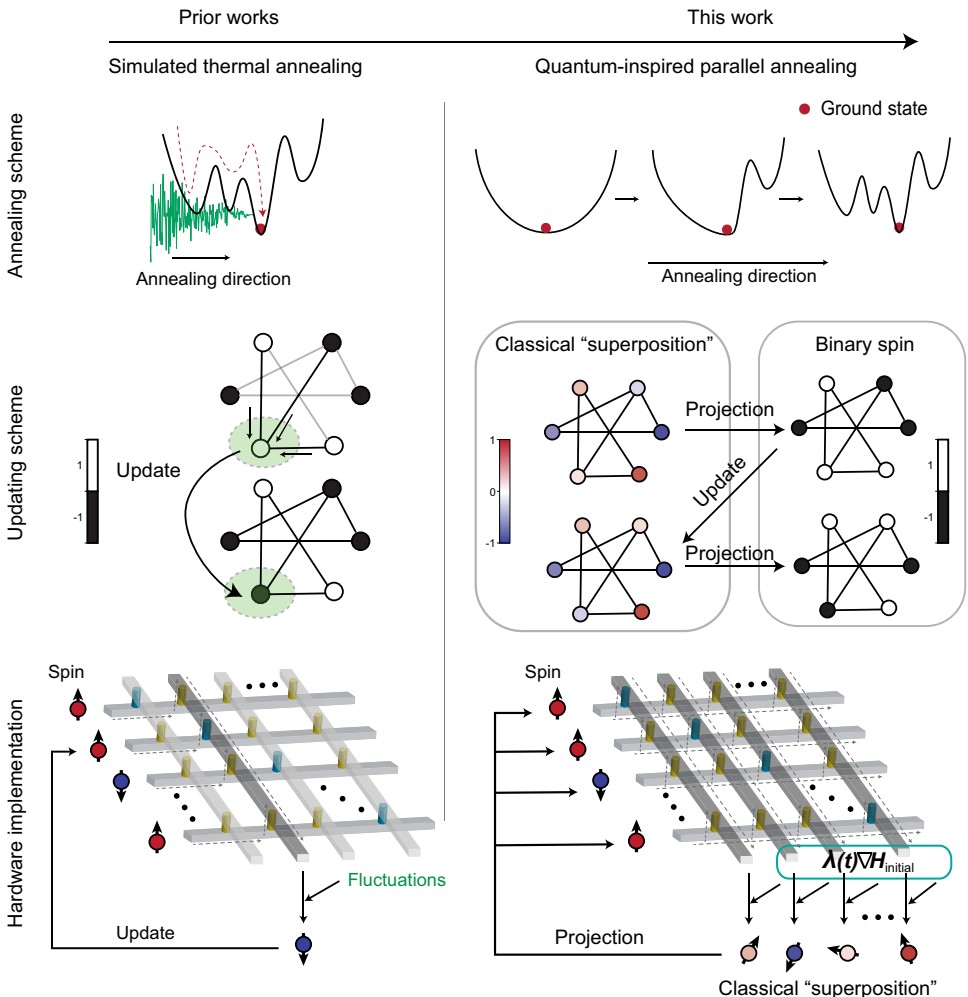

**Fig. 1 | Key properties of quantum-inspired parallel annealing and its difference with simulated thermal annealing.** Simulated thermal annealing, also known as simulated annealing, has a serial updating nature and differs from our quantum-inspired parallel annealing in terms of annealing strategy, updating scheme, and hardware implementation methods. For the annealing strategy, simulated annealing utilizes decreasing noises to get out of the local minimum. Our quantum-inspired parallel annealing is based on the adiabatic shift of the Hamiltonian landscape. For the updating scheme, simulated annealing only supports updating a single spin at each iteration. Our quantum-inspired parallel annealing introduces classical "superposition" as the intermediate spin state. At each iteration, all classical "superpositions" are updated simultaneously by using the gradient calculated by binary spins. For the hardware implementation methods, simulated annealing only utilizes one column of the memristor crossbar array at each update. Our quantum-inspired parallel annealing utilizes the entire array and thus fully unleashes the natural parallelism of memristor crossbar.

Ising Hamiltonian. During the solving process, the Ising couplings between each pair of spins were stored in the memristor crossbar as analog conductance values in an all-to-all manner, which can be used in situ for calculating the gradient of the Ising Hamiltonian in a single step. Therefore, all outputs from the memristor crossbar can be utilized for updating the spins, thus preserving its parallel and analog nature. The key property of our QPA implemented in the analog memristor crossbar and its difference with previous simulated thermal annealing were summarized in Fig. 1.

**Demonstrating QPA in memristor crossbar for Max-Cut problem**
To demonstrate the solving ability of our proposed QPA in memristor crossbar, we chose Max-Cut as the benchmark problem. Max-Cut is a classical and widely studied NP-hard combinatorial optimization problem, that is commonly used to benchmark Ising machines[5,34]. The choice is due to its direct map-ability to the Ising model and significant practical applications, which include integrated circuit routing, computer vision and data clustering, etc. The Max-Cut problem involves dividing all vertices $V$ of a given graph $G(V,E)$ into two sets in a way that cuts the maximum number of edges connecting nodes in different

sets. A detailed explanation of how we mapped this problem to Ising model can be found in the Methods section. In our implementation, the Ising couplings were stored in the memristor crossbar as analog conductance values, which were programmed with an iterative write-and-verify approach. The memristor crossbar was then used for computing gradient by performing matrix multiplication in the analog domain, while the part of spin updating was performed in the digital domain (Fig. 2a). All experiments were conducted on our integrated memristor chip, which consists of multiple 64 × 64 one-transistor one-memristor (1T1M) arrays with necessary peripheral circuits including drivers, multiplexers (MUXs), transimpedance amplifiers (TIAs) and analog-to-digital converters (ADCs) (Fig. 2b). In this demonstration, a personal computer (PC) was used for controlling the chip and updating the spins (Fig. 2a). More details about experiment set-up and electrical characteristics of our memristor device can be found in Figs. S3 and S4.

We started with a 50%-density (50% of edges are connected) unweighted 64-node Max-Cut problem to ensure a fair comparison with recently reported results[16]. The Ising coupling matrix mapped from the problem is shown in Fig. 2c. As only negative Ising coupling

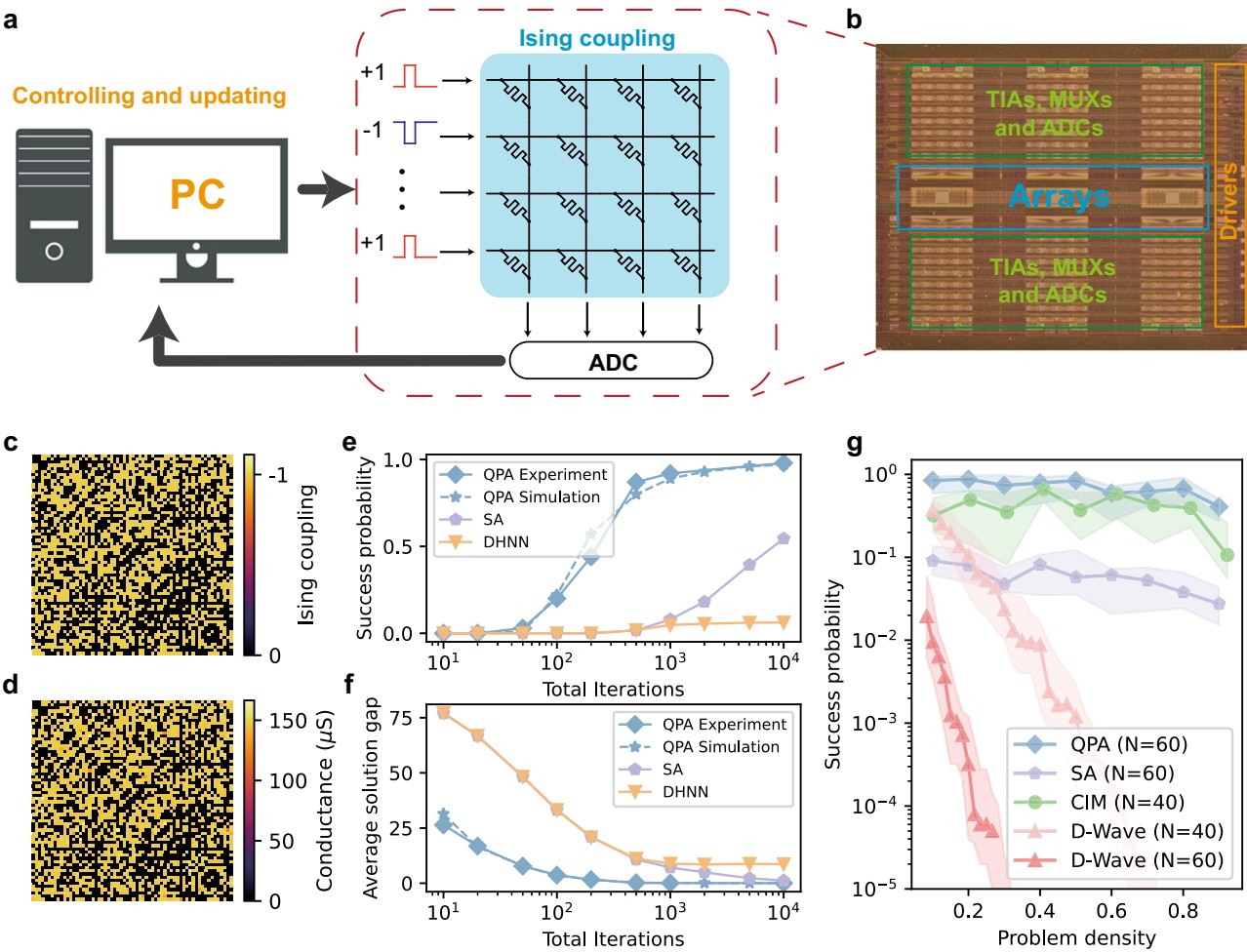

**Fig. 2 | Experimental demonstration of QPA in memristor crossbar array.**
**a** Schematic of experiment set-up. A general-purpose computer controls the integrated memristor chip, which stores the Ising coupling matrix values and computes the gradient in the analog domain, and implements the part of spin updating in the digital domain. **b** Optical image of the integrated memristor chip, which consists of multiple 64 × 64 one-transistor-one-memristor (1T1M) arrays with necessary peripheral circuits including drivers, multiplexers (MUXs), transimpedance amplifiers (TIAs) and analog-to-digital converters (ADCs) **c** Ising coupling matrix of a target 64-node 50%-density Max-Cut problem, and **d** experimental read-out conductance value after programming the Ising coupling onto the memristor crossbar array. "0"s are programmed to target 0 μS, while "−1"s target 150 μS. **e, f** Performance comparison of QPA, SA, and DHNN in terms of **e** success probability and **f** average solution gap (solution gap = optimal maximum

cut − obtained cut) across different total iterations. One iteration represents one attempt to update the spins. The blue dashed line shows the simulation result of QPA, which shows strong agreement with experiment data. Each data point is calculated based on 100 different trials from random initial states. **g** The relationship between success probability and problem density for different techniques including our QPA, SA, CIM, and D-Wave. The shaded area in the plot shows interquartile range (25–75%). For each problem density, 20 randomly generated problem instances are solved. 1000 trials are conducted for each problem instance to calculate the success probability. For QPA and SA, the annealing time is 1000 iterations, which takes 1000 μs if assuming each iteration takes 1 μs. For CIM and D-Wave, the annealing time is also 1000 μs. The results for CIM and D-Wave are replotted from ref. 34. The average solution gap plot for QPA and SA can be seen in Fig. S8.

is needed in Max-Cut problems and conductance value can only be positive, minus Ising coupling (−$J$) was converted into conductance values ranging from 0 to 150 μS. After configuring the conductance values into the hardware, we experimentally read out the resulting conductance matrix, which is shown in Fig. 2d. The agreement between the mapped and experimentally read-out matrices demonstrated the good performance of our device and hardware platform. Once programmed, we experimentally solved the problem using the QPA described earlier without changing the memristor conductance values. In the solving process, the memristor crossbar was used for calculating the gradient, which is basically the vector-matrix multiplication operation, in a single step. While the spin updating including the addition of the gradient of initial Hamiltonian and the updating of classical "superposition" $x$ was performed by the controlling PC in the digital domain at the current demonstration stage and can be further moved to the chip with

customized digital circuits in the futural design. A detailed flow chart can be found in Fig. S5.

After that, the performance of QPA was compared with that of a naïve discrete-time Hopfield neural network (DHNN) and simulated annealing (denoted as SA for short) with different annealing time (total iterations) in Fig. 2e, f. DHNN is a local search algorithm, which serially update the spins towards lower energy states[35]. The SA implementation here involved adding noises with decreasing amplitude to DHNN, equivalent to the decreasing "temperature" in classical SA implementation and is commonly used in memristor-based implementations[16,17,22,36]. The noise level in SA (the standard deviation of the Gaussian distributed noises) changed linearly from 2 to 0 throughout the iterations. Similarly, for our proposed QPA, $\lambda$ changed linearly from 10 to 0, with a fixed step size $\eta = 0.01$ used for the gradient descent part (further discussion about the choice of $\lambda$ and step size $\eta$ can be seen in Fig. S6 and Fig. S7, respectively). The results

demonstrate that the DHNN without annealing has negligible chance of success because of the local search nature, while our proposed method clear gave a better solution in terms of both the success probability of reaching the maximum cut value (ground state) and average cut value with the same total iterations than SA. This is owing to the parallelism of the proposed method, which enabled the system to process and preserve more information in a single iteration. Furthermore, the speed of annealing was affected by the change in the total iteration number in this experiment, as the noise level in SA and the coefficient of initial Hamiltonian $\lambda$ in our QPA changed linearly throughout the iterations. The success probability of SA would eventually reach 100% with sufficient number of iterations, because it has been established that the convergence of SA is guaranteed if the "temperature" changes sufficiently slow[37]. The same is true for adiabatic annealing in quantum systems, where the transfer from the initial Hamiltonian to the Ising Hamiltonian should be sufficient slow (that is obeying the adiabatic evolution rule) to guarantee the system to stay in the ground state[26,38]. This also might be true for our classical version of adiabatic annealing according to the experiment result that success probability continues to grow with the increasing of total iteration number. We expect theoretical proof to follow in the future.

We further investigated how the performance changes with different problem densities and compared the results with those obtained using other techniques (Fig. 2g). Due to the lack of guaranteed convergence without annealing, DHNN was not compared in this analysis. Owing to experimental hardware resource limitations, we used our experimentally-validate memristor crossbar model[39], taking into consideration of non-ideal factors in experiments, which has demonstrated high agreement with the experimental data (Fig. 2e, f) for the experiments we described before. Our findings show that QPA can achieve a higher success probability than all competitive approaches including SA, the coherent Ising machine (CIM), and the D-Wave quantum annealer[34]. In addition, since the memristor crossbar array is naturally dense and thus has all-to-all connectivity, the proposed system can easily handle problems with varying density.

## All-to-all connected weighted Max-Cut solving

One major distinction between our method and other techniques used in Ising machines is the use of memristor crossbar array to store coupling strength, which offers a unique feature: each cross-point can be programmed to an arbitrary conductance state, enabling the representation of arbitrary coupling strength between any spins with all-to-all connectivity. As practical problems usually require more levels of coupling strength, this feature allows systems based on our approach to solve such problems without any additional hardware cost, so as to significantly improve both speed and energy efficiency.

Our previous results already certified that our system can be used to solve highly dense Max-Cut problems. In addition to unweighted ones, we experimentally solved an all-to-all connected weighted Max-Cut problem to fully exploit the analog storage and processing capability of memristor crossbar array. First, a random weighted Max-Cut problem was generated, where the weight of each edge randomly was assigned a random 16-bit integer (from 0 to 65,535) (Fig. 3a). The problem was then mapped to a proper conductance range (0 to 150 $\mu$S) and programmed onto the memristor crossbar array (Fig. 3b). The programming and computing accuracies are illustrated in Figs. 3c and 3d, respectively. The results were compared with SA and DHNN. Figure 3e shows the evolution of the cut value during the solving processing, while the final solution distribution is plotted in Fig. 3f. The evolving process of the classical "superposition" can be observed in Fig. S10. Similar to the unweighted Max-Cut problem, our approach obtains a significantly better solution within a certain number of iterations.

As the weight value became an arbitrary value and introduces more energy states, the problem became harder to solve and resulted

in the poorer performance of SA and DHNN than when they were used to solve unweighted problems, with limited annealing time (Fig. 3e, f). However, our QPA exhibited great performance: 48 out of 100 trials with random initial states finally converge to the true ground state, while neither SA nor DHNN finds the optimal solution even once. Although the variation of memristor inhibited it from being perfectly programmed to a given conductance state, our experimental result confirmed that the current programming accuracy is enough to solve COPs with digital-comparable success probability. Further simulation with different conductance variations was conducted to examine the impact of programming error of memristor crossbar array (Fig. 3g). With a large conductance variation, the success probability of reaching the optimal solution decreases. However, with relatively small conductance variation (0 to 5 $\mu$S), the average solution quality, in terms of both success probability (blue line) and the cut-values (red line), shows negligible degradation, and our experimental conductance variation (about 2.36 $\mu$S as shown in Fig. 3c) is far from the value where the accuracy shows noticeable degradation. In addition, the conductance variation of memristor device can potentially be further reduced with new material stack and denoising techniques to enhance the performance[40] in future systems. For most practical applications scenarios where good-enough sub-optimal solution are accepted, such as 99.5% of the maximum cut value, the analog computing system demonstrates even better tolerance to conductance variation, as shown in almost non-degraded accuracy in Fig. 3g (blue line with pentagon markers), for conductance up to 10 $\mu$S (more than four times our experimental value). This illustrates that the heuristic approach of Ising method is particularly suited for analog computing hardware. Further simulation about the conductance relaxation effect of memristor device can be seen in Fig. S11, which shows that our system still gets acceptable solutions with limited performance degradation after a long retention time.

Furthermore, we investigated how the time-to-solution (TTS) of different approaches scales with the problem size by simulation (Fig. 3h). TTS is defined as the time required to guarantee a 99% success probability of reaching the global optimal solution, typically achieved by performing multiple annealing runs and selecting the best result. Therefore, the TTS can be quantitatively calculated as $TTS = T_{ann} \left\lceil \frac{\lg(1-0.99)}{\lg(1-P)} \right\rceil$, where $T_{ann}$ is the annealing time needed for a single run, and $P$ is the success probability of a single run. The success probability used for calculating TTS can be found in Fig. S12, while the scaling trend of unweighted Max-Cut problem is shown in Fig. S13 along with the exploration of the scaling trend for much larger problems shown in Fig. S14. Further discussion on the hardness of the problem can also be seen in Fig. S15. Notably, for QPA, the results shown here were from simulation considering experimentally available computing error, while for SA, the results were obtained by defect-free simulation, so the actual improvement over SA is larger than what we reported here. Our results in Fig. 3g show that the time complexity of both approaches scales exponentially with the square root of problem size $N$, i.e., $TTS = ae^{b\sqrt{N}}$ where $a$ and $b$ are constants, consistent with previous studies[34,41]. Although our proposed QPA shows a similar scaling trend to SA, the scaling factors are considerably smaller, indicating a more significant scaling advantage. For example, with a problem size of $N = 120$, the TTS of the proposed method is 46× lower than that of SA. Moreover, by simply extrapolating the curves, the speed-up ratio is expected to increase with problem size.

## Traveling salesman problem

In addition to its capability of solving Max-Cut problems, our memristor-based system has the potential to be used for solving other types of combinatorial optimization problems (COPs) due to the universality of the Ising model[4]. To demonstrate this, we chose traveling salesman problem (TSP) as another classical COP benchmark task. TSP involves finding the shortest path that visits each city once and returns

 

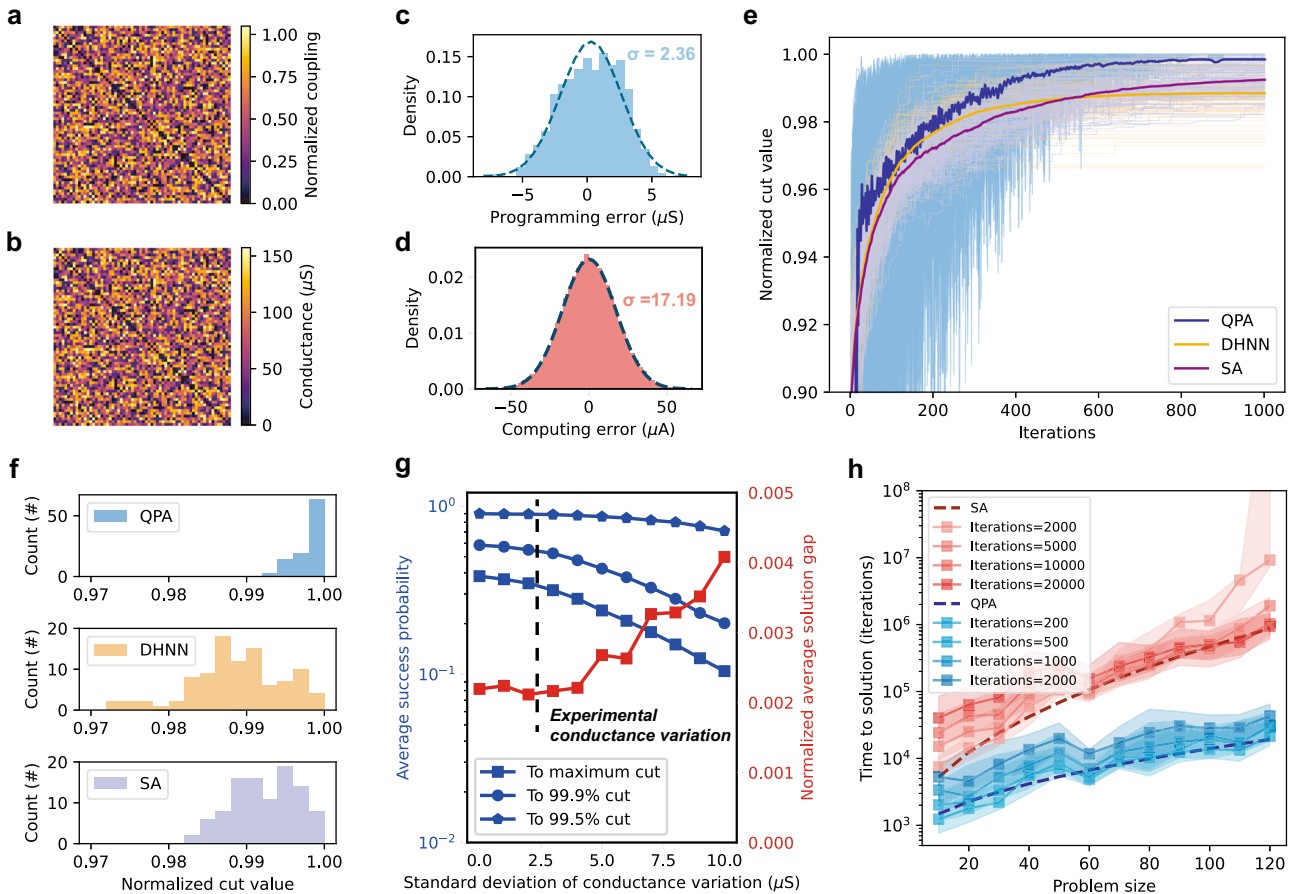

**Fig. 3 | Weighted Max-Cut problem with analog conductance values for Ising coupling. a** Normalized Ising coupling matrix ($-\mathbf{J}/\max(-\mathbf{J})$) of a randomly generated 64-node all-to-all connected Max-Cut problem. Each connection strength is randomly selected from a 16-bit integer (0 to 65,535) **b** Experimental read-out conductance matrix after programming the Ising coupling to the memristor crossbar array, with the maximum coupling strength programmed targeting 150 μS. **c** Distribution of programming error, with the dash line showing the Gaussian distribution fitting with a mean of 0.29 μS and a standard deviation of 2.36 μS. **d** Distribution of computing error of the analog memristor crossbar with 1000 random input vectors. The dashed line indicates a Gaussian distribution with a mean of 0.26 μA and a standard deviation of 17.19 μA. The full output range is about −750 to 750 μA. The direct scatter plot of the experimental result with respect to the expected result can be found in Fig. S9. **e** The calculated cut-values during the solving process with QPA, DHNN, and SA schemes. The goal is to get the maximum

possible cut value. The cut-values shown here are normalized by dividing the obtained cut value by the optimal maximum cut. The light color lines represent 100 different trials from random initial spin configurations, and the dark color lines represent the average cut value. The annealing time for all three methods is 1000 iterations. **f** The corresponding solution cut distribution of three methods. **g** The impact of conductance variations of memristors. For each data point, 20 randomly generated problem instances are solved 1000 times with random initial spin configuration. The blue lines with square, round, and pentagon markers represent the success probability to optimal maximum cut, 99.9% of the maximum cut and 99.5% of the maximum cut, respectively. **h** Scaling trend of time-to-solution (TTS) with problem size. For each problem size, 20 randomly generated problem instances are solved 1000 times with random initial spin configuration. The shaded area in the plot shows interquartile range, the square marker indicates the median, and the dashed line shows the fit of $TTS = ae^{b\sqrt{N}}$.

to the starting city, and has various applications in scheduling and routing problems. To map TSP to the Ising model, we used $(N-1)^2$ spins, where $N$ represents the number of cities, and arranged them to a $(N-1) \times (N-1)$ matrix with rows representing the cities and columns representing the visiting order (Fig. 4a, b). Each row and column should have exactly one spin in the spin-up state to satisfy the constraint that all cities must be visited once and only once. We adopted a binary bit formula (either 0 or 1) for the spin variable, $b_i$, with $b_i \equiv \frac{\sigma_i + 1}{2}$, where $\sigma_i$ is the spin value, which is either −1 or 1, representing spin-down and spin-up states, respectively. The energy is conveniently phrased using this formula, and the coupling matrix $\mathbf{J}$ and bias $\mathbf{h}$ are modified accordingly[4]. The detailed process of mapping TSP to the Ising model is presented in the methods section.

Figure 4c, d shows the ideal Ising coupling strength and experimental read-out conductance matrix after mapping the target problem in Fig. 4a into conductance and configuring it into the memristor hardware (More details about the target problem can be found in Fig. S16). Then, the problem was experimentally solved with the integrated

memristor chip by implementing QPA. And the Ising Hamiltonian evolution was compared to DHNN and SA in Fig. 4e, similar to the Max-Cut problem. The final solution obtained by different solvers after 1000 iterations were compared in Fig. 4f. QPA achieved a much higher solution quality with the same number of total iterations, generating more valid tours and finding tours with a smaller minimum and average distance. This demonstrates the effectiveness of our approach in solving TSP. Further scaling simulation results on solving practical problems in TSPLIB[42] can be seen in Fig. S17. Moreover, the benefits of the analog property of our memristor device enabled a significant reduction in the number of device cells needed to solve TSP. For example, to solve a 10-city TSP, the array size can be reduced from $526 \times 200$ to $81 \times 81$, resulting in a 16.03× deduction in hardware cost compared to the implementation in ref. 22.

It is worth noting that the mapping method from TSP to Ising model may face scalability issues because the required spin number increases with the square of the number of the cities to visit. The problem can be mitigated by advanced clustering techniques, by

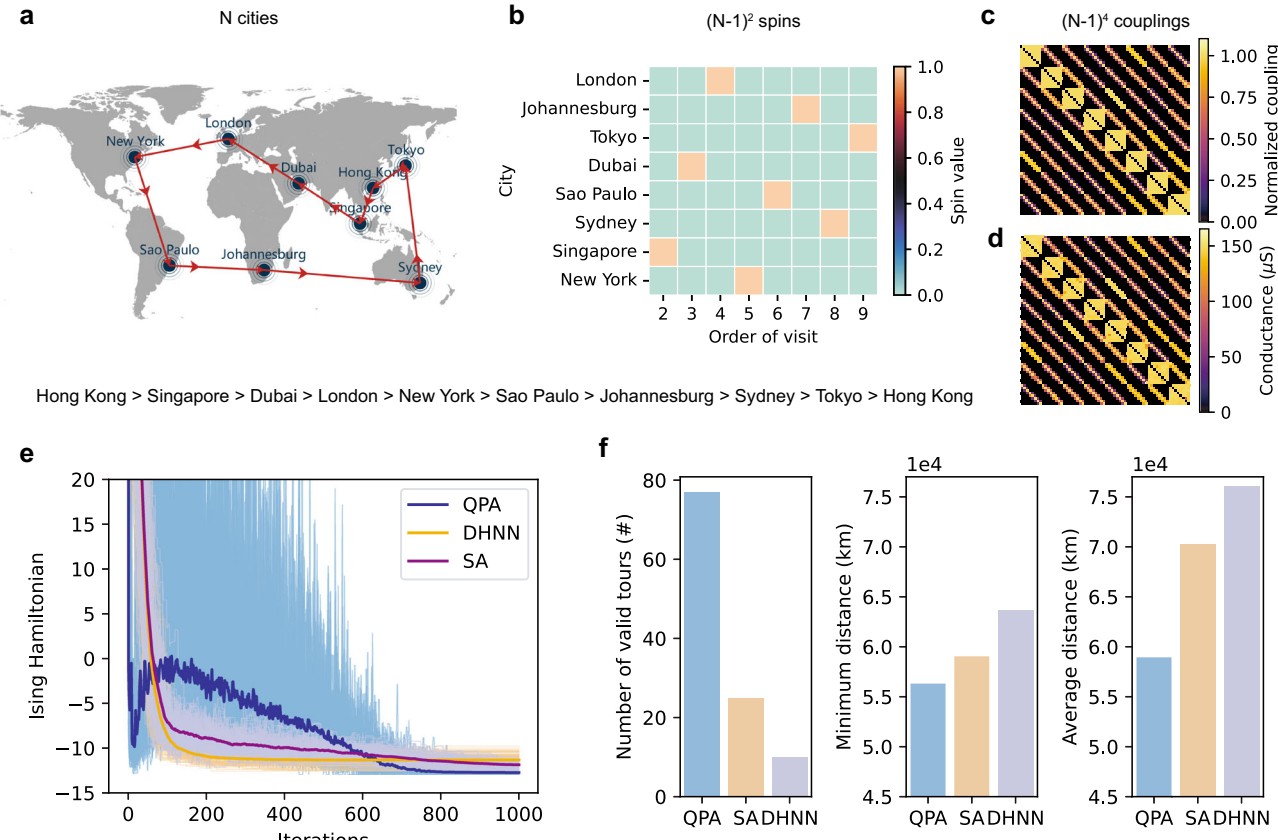

**Fig. 4 | Traveling salesman problem. a** A target $N$-city traveling salesman problem, with solid circles marking the location of Hong Kong, London, Johannesburg, Tokyo, Dubai, Sao Paulo, Sydney, Singapore, Singapore, and New York, respectively. The is to find a route that travels all cities once with the minimum distance. **b** The visiting order represented by an $(N - 1) \times (N - 1)$ matrix, with rows representing the city and columns representing the order of visit. Each row and column can have only one element set to one, representing a valid visiting order. City 1 (Hong Kong in this case) is always chosen as the first city to be visited. The matrix in the plot represents the example valid visiting order shown in (**a**), which is Hong Kong>Singapore>Dubai>London >New York>Sao Paulo>Johannesburg>Sydney>-Tokyo>Hong Kong. **c** Normalized Ising coupling matrix of the target problem, and **d** the experimental read-out conductance value after programming the Ising

coupling to the memristor crossbar array. The maximum coupling strength is programmed targeting 150 μS. **e** The evolution of the Ising Hamiltonian during the solving process. The dark color lin**e**s represent the average cut value over 100 different trails (light-colored lines) from random initial states. 1000 total iterations are used for all three methods. **f** Comparison of solution quality obtained by QPA, SA, and DHNN. Three performance metrics are compared: number of valid tours, minimum distance, and average distance. The number of valid tours represents the number of solutions that obey the constraints of each city should be visited and visited once (the larger the better). Minimum distance refers to the shortest traveling distance among 100 trials, and average distance represents the mean value of the traveling distance of valid tours (the smaller the better).

breaking down a large problem to several levels of smaller problems[43,44]. In this case, the solving speed and the solution quality of small problems can be crucial to the entire large problem. Therefore, the analog property of the device, combined with the parallel quantum-inspired annealing of our memristor-based system for better solution quality, are well suited to such techniques. Moreover, the mixed signal processing of current implementation becomes appealing as it is more compatible to higher level processing that is inevitable for the mapping and clustering.

### Performance benchmark and future directions

Table 1 compares the key properties and performance metrics of various Ising machines, for solving a 100-node dense Max-Cut problem. The key properties include the representation method of spins and couplings, connectivity and precision of couplings, updating and annealing mechanisms, providing basic understanding of each technique. The performance metrics include annealing time, time to solution, power dissipation, and energy efficiency. We compared seven different techniques, including our memristor-based QPA, memristor-based Hopfield networks (mem-HNN)[16], phase-transition nano-oscillators (PTNO) based continuous-time dynamic system[14],

CMOS ring oscillator (ROSC) based Ising system[11], simulated bifurcation machine (SBM) running discrete simulated bifurcation (dSB) on field programmable gate array to represent state-of-the-art digital solver[41], coherent Ising machine (CIM)[8,34], D-Wave 2000Q quantum annealer[6,34]. Details about the estimation breakdown of QPA are shown in Supplementary Table S1. And detailed discussion on benchmark of other technologies can be seen in Supplementary Note S1. We chose 100-node unweighted Max-Cut problems as the benchmark as it is commonly used in other reports, for easier comparison. The QPA implemented on our memristor-based system obtains time to solution of 10.8 μs, which is 2.3× faster than previous state-of-the-art solvers, and obtains energy efficiency of $4.10 \times 10^7$ solutions per second per watt, which is 3.1× greater than previous state-of-the-art solvers. This advantage is primarily attributed to the novel quantum-inspired annealing scheme, which further exploits the parallel, all-to-all connectivity and analog property of memristor crossbar array. It is important to note that the 100-node Max-Cut is not the limitation of memristor-based system, as the state-of-the-art memristor-based in-memory computing macro has $1024 \times 512$ devices in a single bank[45]. With a larger array, the advantage brought by synchronous updating can also be enlarged due to the utilization of higher parallelism.

**Table 1 | Benchmark comparison of QPA implemented in memristor crossbar with other state-of-the-art Ising machines of solving 100-node 50%-density unweighted Max-Cut problem**

| | This work | mem-HNN | PTNO | ROSC | SBM | CIM | D-Wave 2000Q |
|---|---|---|---|---|---|---|---|
| Representation of spins | Digital bits | Digital bits | Oscillator phases | Oscillator phases | Digital bits | Coherent light | Superconducting qubits |
| Representation of couplings | Conductance in memristor crossbar array | Conductance in memristor crossbar array | Capacitance/resistance | Transmission gate | Coupling matrix stored in FPGA | Coupling matrix stored in FPGA | Flux storage |
| Connectivity | All-to-all | All-to-all | All-to-all | Sparse (King's graph) | All-to-all | All-to-all | Sparse (Chimera) |
| Resolution of couplings | Analog | 1 bit | 1 bit | 5 levels | 10 bits | −1 or +1 | 5–6 bits |
| Update mechanism | Synchronous | Hybrid (10-nodes per time step) | Synchronous | Synchronous | Synchronous | Asynchronous | Synchronous |
| Annealing scheme | QPA | Modulating intrinsic noises | Second-harmonic injection lock | – | dSB | Coherent computing | Quantum annealing |
| Annealing time | 400 ns | 1 µs | 10 µs | 50 ns | – | 250 µs | 1 ms |
| Time to solution | 10.8 µs | 25 µs | 30 µs | >23 µs | 29 µs | 2.3 ms | >$10^4$ s (N = 55) |
| Power | 2.3 mW | 16 mW | 2.56 mW | 42 mW | 200 W | >200 W | >25 kW |
| Energy to solution | 24.8 nJ | 400 nJ | 76.8 nJ | 924 nJ | 5.8 mJ | >460 mJ | >250 MJ |
| Solutions per second per watt | $4.03×10^7$ | $2.5×10^6$ | $1.3×10^7$ | $1.08×10^6$ | $1.72×10^2$ | <2.17 | <$4×10^{-9}$ |

The mixed-signal memristor-based approaches, including our memristor-based QPA and mem-HNN, store and compute the coupling term in the analog domain, while implementing spin updating in the digital domain. In contrast, CIM updates spins in the analog domain and calculates coupling terms in digital domain. Considering that the utilization of quantum mechanics of current CIM demonstration remains unclear and that it can be described accurately by classical dynamics[5,46,47], we believe that implementing the coupling term in the analog domain might be more efficient at the current stage before CIM showing quantum advantages. This is because the computing complexity of spin updating is usually $O(N)$, with the possibility of reaching time complexity of $O(1)$ if custom parallel digital logic is implemented. On the other hand, the coupling calculation, which CIM implemented in the digital domain, is a VMM operation with a computing complexity of $O(N^2)$, making it significantly more compute-intensive than spin updating.

In the next phase, it is desired to further push everything to the analog domain, for even higher performance and energy efficiency by eliminating the expensive analog-to-digital conversions. The full-analog approach, however, is more challenging, and requires developing techniques to compensate for parasitic resistance and capacitance, and to reduce analog computing errors[48]. Meanwhile, current mixed-signal design provides more flexibility, and can take advantage of the rapid development of memristor-based AI accelerators, as they share the same data flow[49,50].

## Discussion

In conclusion, we have experimentally demonstrated quantum annealing concept can be applied to classical memristor-based analog in-memory computing hardware, resulting in the memristor-based Ising machine capable of solving combinatorial optimization problems. Our quantum-inspired parallel annealing approach has been validated through various tasks, including 64-node unweighted and weighted Max-Cut and nine-city traveling salesman problems, on our integrated memristor platform. The results show improved efficiency and solution quality. The simulation based on our experimentally validated model further shows increasing advantages on scaled problems of our method compared with serial-updating simulated annealing. This is primarily due to the utilization of massive parallelism provided by memristor crossbar. In addition, our demonstration takes full advantage of the analog conductance states of memristor device, which were configured to represent arbitrary coupling strength between spins. Both experimental and simulation analyses indicate only limited degradation in solution quality with our experimental conductance variation. By implementing the novel quantum-inspired algorithm that fully unleash the potential of our memristor-based hardware, including natural parallelism, analog conductance states and all-to-all connection, our system demonstrates 2.3× speed benefit compared to state-of-the-art mem-HNN and 3.1× energy efficiency compared to state-of-the-art PTNO-based system, and orders of magnitude improvement than Ising machines implemented using other technologies, including oscillator-based, pure digital, CIM and quantum annealer systems.

## Methods

### Memristor integration
We would like to express our gratitude to the team in Hewlett Packard Lab for providing the integrated memristor platform. The memristor devices used in this platform were integrated on top of foundry CMOS chips with a standard 180 nm technology node in an in-house back-end-of-the-line process. The passivation layer of the chip was first removed, which was followed by patterning 2 nm Cr and 10 nm Pt as the bottom electrode. Then, the switching layer of 4–8 nm $TaO_x$ was deposited by reactive sputtering. Finally, 10 nm Ta was sputtering deposited as the top electrode with 10 nm Pt as the

protection layer. More details about the fabrication process can be found in ref. 51.

## Conductance programming

An adaptive and iterative algorithm was employed in this study to adjust the conductance state of the memristors. Before the programming process, the target conductance is generated by scaling the coupling strength to the conductance range of our memristor device (near 0 μS to 150 μS) without any quantization. At the onset of each programming iteration, the conductance of each memristor was measured using a read voltage of 0.2 V, and subsequently compared to the target conductance. If the difference between the measured conductance and the target conductance fell within a predetermined tolerance range (5 μS in this work), the algorithm would halt. However, if the difference exceeded the tolerance range, a SET or RESET pulse was applied to the device to increase or decrease the conductance and approach the desired target. Throughout the iterations, the SET and RESET voltages, as well as the gate voltages, were adaptively increased if the algorithm consistently pursued a specific direction of change.

## Clipping and momentum

During the solving process, after each updating iteration, the analog proxy spin $x$ is clipped between −1 and 1, using the following equation:

$$\text{clip}(x, -1, 1) = \max(-1, \min(1, x)) \tag{6}$$

This clipping is a common practice in Binary Neural Network (BiNN) training to prevent the parameter value from growing infinitely, so that a slight change in the value will not have any effect on the result of binarized parameter[32,52]. And to further increase the convergence speed, a momentum gradient descent is adopted[53]:

$$m(t+1) = \beta^* m(t) - \eta * \text{gradient} \tag{7}$$

$$x(t+1) = x(t) + m(t+1) \tag{8}$$

where $\beta$ to be the momentum constant, which is set to 0.99 through this paper for simplicity. The momentum $m$ is clipped between −1 and +1 to be prohibited from explosion.

## Mapping Max-Cut to Ising model

The Max-Cut problem aims to divide all nodes into two subsets. To mathematically represent the problem, we use $s_i = 1$, if the $i$th node is in one subset and $s_i = -1$, if $i$th node is in the other subset. $A$ is the adjacency matrix with elements defined as $A_{ij} = 0$, if there is no edge between ith and $j$th nodes and $A_{ij} = 1$, if there is an edge. Then the cut number can be expressed as,

$$Cut = \frac{1}{2} \sum_{i<j} A_{ij} \left(1 - s_i s_j\right) \tag{9}$$

As $\sum_{i<j} A_{i,j}$ is a problem-defined constant, maximizing cut number is equivalent to minimizing $-\sum_{i<j}(-A_{i,j})s_i s_j$ and thus the problem can be mapped to the Ising Hamiltonian by setting $J = -A$ without the need of local field term $h$.

## Mapping TSP to Ising model

The mapping method implemented in this paper is improved from the TSP encoding scheme described in ref. 4. To map an $N$ city TSP problem, $N$ spins are required. Each spin is represented in binary bit form (either 0 or 1) and is denoted as $b_{v,j}$, where $v$ represents the city and $j$ represents the visiting order. The Ising Hamiltonian can be defined by two parts:

$$H_A = A \sum_{v=1}^{N} \left(1 - \sum_{j=1}^{N} b_{v,j}\right)^2 + A \sum_{j=1}^{N} \left(1 - \sum_{v=1}^{N} b_{v,j}\right)^2 \tag{10}$$

$$H_B = \frac{B}{2} \sum_{u,v=1}^{N} D_{uv} \sum_{j=1}^{N} \left(b_{u,j} b_{v,j-1} + b_{u,j} b_{v,j+1}\right) \tag{11}$$

$$H = H_A + H_B \tag{12}$$

$H_A$ imposes constraints to ensure that each city is visited and is visited only once. $H_B$ models the summation of the traveling distance of two adjacent visited city, where $D_{uv}$ is the traveling distance between $u$ city and $v$ city. Since each traveling distance is calculated twice, the summation is halved. A and B are coefficients of $H_A$ and $H_B$, which determines the contribution of the constrains and traveling distance to the overall Hamiltonian. To balance the validity of the solution and the quality of the solution, B is set to 1 and A is set to max $(D_{uv})$ throughout this paper[4]. Moreover, city 1 is always chosen as the first city to visit and thus reduce the required spin number to $(N-1)^2$. This can be understood by fixing the $2N-1$ spins to represent city 1 and visiting order 1. This has a constant effect on other spins and only modifies the local field term $h$, which is added in digital domain in our implementation, and does not change the coupling strength between remaining $(N-1)^2$ spins.

## Problem instance generation and optimal solution

All Max-Cut problem instances used in this paper are generated by the *random* module of *numpy* in python environments with default random seeds. For unweighted Max-Cut problems, a predefined number of "0"s and "1"s are given first and shuffled by *numpy.random.shuffle* function to ensure a specific density. For weighted Max-Cut problems, the connection strength is assigned using the *numpy.random.randint* function by randomly selecting a 16-bit integer (ranging from 0 to 65,535). Since the running time of exhaustive search exceeds $10^5$ s for a single problem with problem size $N > 50$ and continues to scale exponentially with problem size[54]. The optimal solution used in this paper is obtained by running SA for enough long time, i.e., $5N \log(N)$ updating cycles (one updating cycle means updating all spins once and corresponds to $N^2$ iterations), with 1000 trials for each problem and selecting the best solution among them, to ensure a high confidence of reaching the true optimal solution[54].

## Data availability

The data that support the findings of this study are provided within the main text and Supplementary Information. Data related to the study can also be made available from the corresponding author upon request. Preliminary results from this study have been reported in the conference proceedings of the 2022 IEEE International Electron Devices Meeting (IEDM)[25].

## Code availability

A demonstration code of quantum-inspired parallel annealing algorithm is available at the GitHub repository: https://github.com/kyshan/QPA. Additional codes related to this study are available from the corresponding author upon reasonable request.

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

## Acknowledgements
The authors would like to thank Xia Sheng, John Paul Strachan, Giacomo Pedretti, Jim Ignowski from Hewlett Packard Labs for providing the integrated memristor crossbar platform. This work was supported in part by the Early Career Scheme from the Research Grant Council of Hong Kong SAR under Grant 27210321; in part by the NSFC Excellent Young Scientist Fund (Hong Kong and Macau) under Grant 62122005; in part by the Croucher Foundation; in part by the Mainland-Hong Kong Joint Fund Scheme (MHKJFS) under Project MHP/066/20; and in part by the ACCESS—AI Chip Center for Emerging Smart Systems, sponsored by InnoHK funding, Hong Kong SAR.

## Author contributions
M.J. and C.L. conceived the idea and M.J. performed experiments. M.J., K.S., and C.H. performed the simulation. C.L. supervised the project. M.J. and C.L. wrote the manuscript with input from all authors.

## Competing interests
The authors declare no competing interests.
