## [Peer Review File · Nature Communications]

REVIEWER COMMENTS

Reviewer #1 (Remarks to the Author):

Jiang and colleagues have demonstrated the use of quantum-inspired tricks in classical computers to solve optimization problems. The paper is well written, the ideas are solid, there is realistic experimental demonstration and also experimentally validated simulations for benchmarking. While there is prior work talking about some of the overall ideas here, the specific tricks in this paper are new. In particular, the idea of using proxy spins and latent weights for optimization is new (as far as I know), and is the intellectual centerpiece of this article. The paper very well rounded, so I have no major issues. I would like to see this work published as soon as possible. A few minor suggestions:

1. It is unclear what results are obtained by simulations only and which are by experiments. Please make this distinction.
2. the size scaling studies are unclear. I would have liked to see a plot of problem size vs accuracy for much larger problem (which is likely what you have done in Fig. S11), but the sizes of the problems there are unclear and the trend of accuracy vs. size is also unclear.
3. All your problems are computationally hard, which is the main selling point of these techniques. However, it is unclear as to how the techniques fare with differing levels of hardness. Again, in your simulations and experiments, this issue is unclear. You have 20 randomly generated problems in many of your trend graphs, and the interquartile range is shown. But what does the full range look like - in other words, are there extreme outliers? What does this full range look like as you increase the number of randomly generated problems (i.e., can you identify the effects of harder problems)?
4. Clean up some of the figures. For example, in Fig. 1, the difference between "superposition/multi-shaded" and "binary/two-shaded" is not visually obvious.

Reviewer #2 (Remarks to the Author):

The manuscript by Li et al. reports a modified quantum-inspired parallel annealing (QPA) algorithm based on the adiabatic annealing approach to solve the combinatorial optimization problems (COPs). This technique uses gradient descent method to update the spin to unleash the potential of parallelism

of MVM crossbar and analog resistive switching of memristor. Typical problems of Max-Cut and traveling salesman are experimentally demonstrated based on a 64×64 1T1M array. This work is a comprehensive summary of a new memristor-based solution for COPs, and shows speed and energy benefits compared to previous works. Overall, the topic is interesting, and the paper is well written. However, more clarifications are needed to fully convince the readers. Detailed technical comments are elaborated as follows.

1) First of all, the fabrication of the 1T1M array shall be described with sufficient details about the memristor device structure.

2) How much computation is accomplished by the 1T1M array is unclear. Please summarize and clarify how and where each part of the calculations is conducted. This is important to show the role of memristor crossbars in such COPs solver.

3) More details about the programming process shall be provided. For example, what's the programming margin? Is there any quantization? What's the bit resolution for the ADCs?

4) According to Fig. 3g, the success probability of realizing the maximum cut is highly sensitive to the precision of the memristor conductance. Would digital versions of the MVM accelerator be more suitable for the QPA approach in COP solving? In practical applications, what are the challenges for solving large-scale COPs based on the memristor technology?

5) Also, the conductance variation included in Fig. 3g only takes the programming accuracy into account, while memristors usually suffer from considerable relaxation effect (i.e., post-programming conductance instability). Is the relaxation effect evaluated for the memristor array? How does it affect the success probability in the demonstration?

6) Also, more electrical measurement data from the 1T1M array shall be provided, such as the endurance and analog resistive switching characteristics with a series of set/reset pulses.

7) In the benchmark in Table 1, it appears that the performance of the proposed memristor-based QPA is obtained by estimation on future implementations with 16nm technology node. Since a functional memristor macro is already fabricated in this work, please also show the benchmark results based on the experimentally measured data.

8) The comparison of solution gap should be included in Fig. 2g.

9) It is recommended to add the units of computing error in Fig. 3d, and plot the curve of the measured MVM outputs with respect to the ideal MVM results directly.

We thank all the reviewers for their precious time and constructive comments on our manuscript. Following the reviewer's suggestions, we have revised both the main text and the supplementary information (SI) accordingly. The major changes we made are summarized as follows:

1. We added a discussion on the scaling trend for larger problems, as shown in *Fig. S14* in the revised SI, and a discussion on the hardness of the problem, as shown in *Fig. S15* in the revised SI.
2. We added details about device fabrication and the conductance programming technique as *Method* on *Page 21* of the revised main text.
3. We conducted further analyses on the conductance relaxation effect of the memristor on the performance of the solver, as shown in *Fig. S11* in the revised SI.
4. We added the analog resistive switching characteristics with a series of set/reset pulses, as shown in *Fig. S4* in the revised SI.

In the following point-to-point response, the original reviewers' comments are in black font, our responses are in blue font. Changes made in the revised main text and SI are highlighted in blue italic font.

Response to Reviewers' comments:

Reviewer #1 (Remarks to the Author):

Jiang and colleagues have demonstrated the use of quantum-inspired tricks in classical computers to solve optimization problems. The paper is well written, the ideas are solid, there is realistic experimental demonstration and also experimentally validated simulations for benchmarking. While there is prior work talking about some of the overall ideas here, the specific tricks in this paper are new. In particular, the idea of using proxy spins and latent weights for optimization is new (as far as I know), and is the intellectual centerpiece of this article. The paper very well rounded, so I have no major issues. I would like to see this work published as soon as possible. A few minor suggestions:

Response: We thank the reviewer for the positive and encouraging feedback. We are grateful for the recognition of the novelty in our application of proxy spins for optimization. Additionally, we highly appreciate the reviewer's suggestions to improve our manuscript.

1. It is unclear what results are obtained by simulations only and which are by experiments. Please make this distinction.

Response: We agree that more elucidation is necessary to highlight the extent of our experimental efforts, and we thank the reviewer for pointing it out. Consequently, we have made changes in both the revised main text and the SI to clarify the distinction between the results obtained from simulations and those from experiments.

1. We revised the following sentence in *Line 173* of the main text:
"Once programmed, we *experimentally* solved the problems using the QPA method described earlier without changing the memristor conductance values."
2. We revised the following sentence in *Line 274* of the main text:
"Furthermore, we investigated how the time-to-solution (TTS) of different approaches scales with the problem size by *simulation* (*Fig. 3h*).
3. In the caption of *Fig. S12* and *Fig. S13* in SI, we added:
"The result for quantum-inspired parallel annealing is obtained by simulation with *experiment-validated memristor model*."

2. the size scaling studies are unclear. I would have liked to see a plot of problem size vs accuracy for much larger problem (which is likely what you have done in Fig. S11), but the sizes of the problems there are unclear and the trend of accuracy vs. size is also unclear.

Response: We thank the reviewer for the constructive comments on improving our size scaling studies. To address this, we have solved and benchmarked a significantly larger Max-Cut problem, with $N=800$ and $N=2000$, using data from a public dataset named as “Gset”.

The results are summarized in Fig. S14 in the revised SI, which shows that the scaling trend of solving larger problems generally aligns with the fitness curve found in Fig. S13(c), with certain deviations.

There are two primary explanations for these deviations. Firstly, the problem instances in the Gset are considerably sparser than those we solved in Fig. S13 (density=0.06 for $N=800$, density=0.01 for $N=2000$, in contrast to density=0.5 for $N=10-100$). Based on our observations in Fig. 2(g) from the main text, these sparser problems are generally easier to solve. Secondly, the simulation conducted in Fig. S13 considered the variation of the memristor device, while the simulation here is defect-free, which results in higher performance (i.e., smaller time-to-solution).”

“Figure S14. Verification of the scaling trend for larger problems with defect-free simulation. (a) The result of solving $N=800$ and $N=2000$ problems with random graphs from a MAX-CUT benchmark dataset, Gset (<https://web.stanford.edu/~yyye/yyye/Gset/>). For each problem instance, 1000 trials are conducted. The number in the bracket denotes the success probability of finding the obtained best cut. (b) Further scaling of the trend in Fig. S12(c) to $N=2000$. One major challenge for benchmarking the performance of large-size problems is to find exact solutions for them. As brute-of-force search becomes impossible for problems with $N>60$ even with the most advanced digital hardware due to the exponential explosion nature of the problem. Thus, Gset problems drawn from a public problem set are used for benchmarking. It can be seen that the scaling trend of solving larger problems is basically consistent with the fitness curve found in Fig. S12(c) with some deviations. There are two major reasons for the deviation. Firstly, problem instances in the Gset are much sparser than problems we solved in Fig. S12 (density=0.06 for $N=800$, density=0.01 for $N=2000$ compared with density=0.5 for $N=10-$

100) and are generally easier to be solved according to our finding in Fig. 2(g) of the main text. Secondly, the simulation conducted in Fig. S12 considered the variation of the memristor device, while the simulation here is defect-free, which results in higher performance.”

Additionally, we have provided clarification regarding the problem size used in Fig. S11 of the original SI (corresponds to Fig. S17 in the revised SI). The following changes have been made in the caption of Fig. S17 in the revised SI:

“Figure S17. Simulation result of solving larger problems in TSPLIB. a Result of solving burma14 problem with 14 cities b Result of solving ulysses16 problem with 16 cities c Result of solving gr17 problem with 17 cities.”

3. All your problems are computationally hard, which is the main selling point of these techniques. However, it is unclear as to how the techniques fare with differing levels of hardness. Again, in your simulations and experiments, this issue is unclear. You have 20 randomly generated problems in many of your trend graphs, and the interquartile range is shown. But what does the full range look like - in other words, are there extreme outliers? What does this full range look like as you increase the number of randomly generated problems (i.e., can you identify the effects of harder problems)?

Response: We thank the reviewer for the valuable comments on the hardness of the problem. To show statistical data including potential outliers and to attain a more comprehensive understanding of the impact of the problem hardness, we conducted expanded simulations. This involved solving 1000 instances of a 60-node 50%-density unweight Max-Cut problems, following which, we plotted the histograms of the success probability and compared the results obtained by different algorithms.

Results and conclusions are summarized in the newly added Fig. S15 in the revised SI, which is also included below for your convenience. From the result, it becomes clear that our method (panels a and b in the first column) has a significantly lower probability of encountering extreme outliers when compared to other methods, including the one based on continuous-time dynamics which only involve the analog spin variable in the solving process. The feature of not separating problems into hard ones and easy ones enables our method to have the advantage of adaptivity in solving broad-spectrum of problems.

“Figure S15. **Exploration of the effect of the hardness of the problem with defect-free simulation.** The success probability distribution of 1000 50%-density unweighted Max-Cut problem instances solved by (a)-(b) Original QPA with (a) 1000 iterations and (b) 10000 iterations. (c)-(d) QPA without discretizing the proxy spin value when calculating the gradient with (d) 1000 iterations and (e) 10000 iterations. (e)-(f) SA with (e) 10000 iterations and (f) 100000 iterations. For each problem instance, 100 random initialized trials are conducted to calculate the success probability. It can be seen that the histogram for QPA without STE is bimodal, meaning that it separates the problem instances into two groups, hard ones and easy ones. The algorithm almost always succeeds on easy problems and fails on hard problems. This kind of bimodal distribution is similar to the result obtained by the D-Wave quantum system and classical dynamics [7]. However, for original QPA and SA, the distribution is relatively unimodal. Further increasing of the total iteration number drives the distribution of QPA without STE more bimodal (The number of both easy and hard problems increases, while the number of problems in the middle decreases) but does not significantly improve the overall success probability. However, for both original QPA and SA, increased iterations cause the unimodal distribution to shift right, which stands for overall improved performance. The detailed mechanism behind the clear separation of hard and easy problems remains unclear. Here, we have a preliminary intuitive explanation. For QPA without STE, the whole solving process is based on the evolution of the analog spin value, meaning that the system tends to evolve to the ground state in the “analog” solution space, which might deviate from the true ground state in the binary solution space for some problem instances. Increasing iterations increases the ability to find the ground state in the “analog” solution space but does not help with finding the true ground state of binary spin. However, the introduction of STE in our QPA utilizes the binary spin value for the gradient calculation, which involves the true binary spin in the solving process and thus corrects the mismatch caused by the analog variable and leads to the true ground state. This might also explain why our method behaves better than those based on continuous-time dynamics such as optical and electronic oscillator-based systems.”

Reference:

“7. Boixo, S., et al., Evidence for quantum annealing with more than one hundred qubits. *Nature Physics*, 2014. 10(3): p. 218-224.”

4. Clean up some of the figures. For example, in Fig. 1, the difference between "superposition/multi-shaded" and "binary/two-shaded" is not visually obvious.

Response: We thank the reviewer for the suggestion. The color used in Fig. 1 has been modified to be more obvious. And a “two-state” black-and-white color bar and “multi-state” gradient color bar are added to strengthen the difference between “binary/two-shaded” and “superposition/multi-shaded”. Moreover, more descriptions have been added to the figure caption to help the audience to understand the figure.

The following changes have been made to Fig. 1 in the revised main text:

“Figure 1. Key properties of quantum-inspired parallel annealing and its difference with simulated thermal annealing. Simulated thermal annealing, also known as simulated annealing, has a serial updating nature and differs from our quantum-inspired parallel annealing in terms of annealing strategy, updating scheme and hardware implementation methods. For the annealing strategy, simulated annealing utilizes decreasing noises to get out of the local minimum. Our quantum-inspired parallel annealing is based on the adiabatic shift of the Hamiltonian landscape. For the updating scheme, simulated annealing only supports updating a single spin at each iteration. Our quantum-inspired parallel annealing introduces classical “superposition” as the intermediate spin state. At each iteration, all classical “superpositions” are updated simultaneously by using the gradient calculated by binary spins. For the hardware implementation methods, simulated annealing only utilizes one column of the memristor crossbar array at each update. Our quantum-inspired parallel annealing utilizes the entire array and thus fully unleashes the natural parallelism of memristor crossbar.”

Reviewer #2 (Remarks to the Author):

The manuscript by Li et al. reports a modified quantum-inspired parallel annealing (QPA) algorithm based on the adiabatic annealing approach to solve the combinatorial optimization problems (COPs). This technique uses gradient descent method to update the spin to unleash the potential of parallelism of MVM crossbar and analog resistive switching of memristor. Typical problems of Max-Cut and traveling salesman are experimentally demonstrated based on a 64×64 1T1M array. This work is a comprehensive summary of a new memristor-based solution for COPs, and shows speed and energy benefits compared to previous works. Overall, the topic is interesting, and the paper is well written. However, more clarifications are needed to fully convince the readers. Detailed technical comments are elaborated as follows.

Response: We thank the reviewer for the positive comments and suggestions on how to improve this work.

1) First of all, the fabrication of the 1T1M array shall be described with sufficient details about the memristor device structure.

Response: We thank the reviewer for the suggestion. Fabrication details of memristor integration have been added to the *Method* of revised main text starting with *Line 430*.

“Memristor integration

We would like to express our gratitude to the team in Hewlett Packard Lab for providing the integrated memristor platform. The memristor devices used in this platform were integrated on top of foundry CMOS chips with a standard 180 nm technology node in an in-house back-end-of-the-line process. The passivation layer of the chip was first removed, which was followed by patterning 2 nm Cr and 10 nm Pt as the bottom electrode. Then, the switching layer of 4-8 nm TaO_x was deposited by reactive sputtering. Finally, 10 nm Ta was sputtering deposited as the top electrode with 10 nm Pt as the protection layer. More details about the fabrication process can be found in Ref. [51].”

Reference:

“51. Sheng, X., et al., Low-Conductance and Multilevel CMOS-Integrated Nanoscale Oxide Memristors. Advanced Electronic Materials, 2019. 5(9).”

2) How much computation is accomplished by the 1T1M array is unclear. Please summarize and clarify how and where each part of the calculations is conducted. This is important to show the role of memristor crossbars in such COPs solver.

Response: We thank the reviewer for the comment. More descriptions have been added to *Line 174* of the revised main text to clarify the function of memristor crossbar.

“In the solving process, the memristor crossbar was used for calculating the gradient, which is basically the vector-matrix multiplication operation, in a single step. While the spin updating including the addition of the gradient of initial Hamiltonian and the updating of classical “superposition” x was performed by the controlling PC in the digital domain at the current demonstration stage and can be further moved to the chip with customized digital circuits in the futural design. A detailed flow chart of the whole solving process can be found in Fig. S5.”

Moreover, a flow chart with green color boxes representing steps conducted experimentally by memristor crossbar array has been added as *Fig. S5* in the revised SI to better illustrate the whole solving process and the role memristor crossbars played.

“Figure S5. Flow chart of the proposed memristor-based Ising solver. Steps in green boxes were conducted experimentally in the analogue memristor crossbar array, while steps in yellow boxes were computing steps in the digital domain, which can be integrated onto the chip in the future.”

3) More details about the programming process shall be provided. For example, what’s the programming margin? Is there any quantization? What’s the bit resolution for the ADCs?

Response: We thank the reviewer for the suggestion. To give a quick answer to the questions, our programming margin is set to 5 μS . There is no quantization in the programming process. The bit resolution for the ADCs is 10-bit, with the option to enable only 5-bit for improved energy benefit. Following the reviewer's suggestion, more details about our programming techniques have been added to the *Method* of revised main text starting with *Line 438*.

“Conductance programming

An adaptive and iterative algorithm was employed in this study to adjust the conductance state of the memristors. Before the programming process, the target conductance is generated by scaling the coupling strength to the conductance range of our memristor device (near 0 μS to 150 μS) without any quantization. At the onset of each programming iteration, the conductance of each memristor was measured using a read voltage of 0.2V, and subsequently compared to the target conductance. If the difference between the measured conductance and the target conductance fell within a predetermined tolerance range (5 μS in this work), the algorithm would halt. However, if the difference exceeded the tolerance range, a SET or RESET pulse was applied to the device to increase or decrease the conductance and approach the desired target. Throughout the iterations, the SET and RESET voltages, as well as the gate voltages, were adaptively increased if the algorithm consistently pursued a specific direction of change.”

4) According to Fig. 3g, the success probability of realizing the maximum cut is highly sensitive to the precision of the memristor conductance. Would digital versions of the MVM accelerator be more suitable for the QPA approach in COP solving? In practical applications, what are the challenges for solving large-scale COPs based on the memristor technology?

Response: We appreciate the reviewer's insightful comment regarding the influence of memristor conductance precision. This is indeed a common concern for any analog computing hardware, including those for accelerating neural networks. The major reason for preferring analog hardware over its digital counterpart is the significant improvement in energy efficiency. In our case, we observe an improvement of five orders of magnitude from the analog accelerator compared to digital ones, such as FPGA, as estimated in *Table 1* of the main text.

However, we acknowledge that all analog computing platforms inevitably exhibit computing errors resulting from various sources, including memristor conductance variation. The key point here is that such computing is more suitable for applications where precise results are not strictly required. For example, the widely explored neural network, and the COP solver we proposed in this work fit perfectly under this criterion. As a sub-optimal solution is usually acceptable (as clearly demonstrated in *Fig. 3g*), a solution that is close to the most optimal one often consumes far fewer computing resources. This is why heuristic approaches (including our QPA) have gained popularity in solving COPs. Although heuristic methods cannot guarantee to always find the most optimal solution, they can consistently return a good enough solution. In this sense, our analog memristor-based solver is perfectly suitable for heuristic approaches, as its performance in finding approximate, good-enough sub-optimal solutions is almost unaffected by the currently available conductance variation level, according to our result in *Fig. 3g* of the main text.

Meanwhile, efforts are ongoing to improve the precision of analog computing based on emerging analog devices at different levels. At the device level, recent results show encouraging progress in achieving 2048 conductance levels in a single memristor device [R1], which is equivalent to 11-bit resolution. This promises significant improvement in analog computing applications, including COPs. From the architecture side, recently proposed analog slicing techniques can significantly improve programming accuracy with limited overhead [R2].

As for the question about the challenges associated with solving large-scale COPs based on memristor technology in practical applications, while our study showcases the feasibility and potential of implementing a quantum-inspired algorithm in memristor crossbar for solving COP, we recognize that there are still challenges in solving large-scale COPs with practical applications. These challenges are largely consistent with those encountered in applying memristors for large-scale artificial neural networks. Some of the key challenges include: 1. Reliability: Due to the imperfections and variations in fabrication, some memristor cells might not repose correctly and get stuck at a certain state or have limited endurance cycles, which largely harms the computing accuracy. Although such an issue can be mitigated by some error-correcting schemes to some extent [R3], it remains challenging to fully eliminate the problem when solving large-scale problems. 2. Scale-up: The scaling problem has two different levels. At the circuit level, the problem is how to increase the crossbar array size in a single tile and keeps the computing accuracy at the same time regarding a series of challenges including, IR drops on the wire, parasitic capacitance, noises introduced by memristor devices and peripheral circuit, ADC design for larger sensing range, heat dissipation with larger current, etc. At the architecture level, the problem is how to coordinate crossbar arrays in different tiles to conduct computation in a parallel fashion with limited overhead.

Reference:

R1. Rao, M., et al., Thousands of conductance levels in memristors integrated on CMOS. *Nature*, 2023. 615(7954): p. 823-829.

R2. Pedretti, G., et al., Redundancy and Analog Slicing for Precise In-Memory Machine Learning—Part I: Programming Techniques. IEEE Transactions on Electron Devices, 2021. 68(9): p. 4373-4378.

R3. Li, C., et al. Analog error correcting codes for defect tolerant matrix multiplication in crossbars. in 2020 IEEE International Electron Devices Meeting (IEDM). 2020. IEEE.

5) Also, the conductance variation included in Fig. 3g only takes the programming accuracy into account, while memristors usually suffer from considerable relaxation effect (i.e., post-programming conductance instability). Is the relaxation effect evaluated for the memristor array? How does it affect the success probability in the demonstration?

Response: We thank the reviewer for the valuable comment. We agree that the relaxation effect of the memristor device might be crucial to the performance. Therefore, we conducted further simulation with an experiment-extracted model to explore the relaxation effect on the performance of our solver. First, we conducted retention experiments to show the overall changing trend of the standard deviation of the conductance variation (read-out conductance – expected conductance) versus time, based on 4,096 devices in a 64×64 array. Then we utilized the experiment-found conductance variation level for simulation and the results show that the success probability and average solution quality slightly degrades over time due to the conductance relaxation effect. Still, the degradation is acceptable with a relatively long retention time (up to 10^5 s). And our system maintains the capacity to provide sub-optimal but good enough solutions (e.g., 99.5% of the maximum cut), proving that our solution is highly practical for certain real-world applications.

The results and conclusions are summarized and added to Fig. S11 in the revised SI.

“Figure S11. Exploration of the effect of conductance relaxation on the performance by simulation with the experiment-extracted model. (a) Experimentally measured relationship between the standard deviation of the conductance variation of the whole array and time. The measuring condition is the same as Fig. S4(c). With the increasing of retention time, the conductance variation of the memristor device gradually grows. (b) Change of average success probability with regard to the retention time. (c) Change of normalized average solution gap with regard to the retention time. For (b) and (c), the result is obtained by simulation with the experimentally measured conductance variation in (a). It can be seen that though with a long power-down period, the non-volatile property of memristor enables our system to get acceptable solutions with limited performance degradation.”

6) Also, more electrical measurement data from the 1T1M array shall be provided, such as the endurance and analog resistive switching characteristics with a series of set/reset pulses.

Response: We thank the reviewer for the suggestion. In response to the constructive comment, we added more electrical measurements of the 1T1M array, including the analog resistive switching characteristics with a series of set/reset pulses, which have been added to Figure S4. as panel d in the revised SI, as shown below:

“Figure S4. Performance of memristor device in our integrated hardware. a DC I-V measurement of 100 Set and Reset cycles demonstrating small cycle-to-cycle variations. The average value is shown by the dark line. **b** Linearity test of different conductance states, demonstrating excellent I-V linearity. **c** Retention test after dividing the entire 64×64 array into 16 subarrays, with each subarray programmed to target a conductance of 0 to 150 μS with a step size of 10 μS . The mean value is shown by the dark line, while the shaded area indicates the interquartile range. **d** Conductance tuning by a set of SET and RESET pulses. 256 SET pulses with amplitude linearly increasing from 0.6 V to 0.7 V were applied to the device followed by 256 RESET pulses with amplitude linearly decreasing from -0.5 V to -0.65 V. The pulse width for both SET and RESET pulses is 10 ns.”

The endurance performance test of the devices with the same material stack in 1T1M cells was conducted previously, which demonstrated more than 1 million programming cycles [R4], the result of which is also reproduced here for the convenience of reference. For the integrated array with the same device, we have already repeatably stressed the chip for more than 10,000 write pulses, so we are confident about the endurance performance. But, considering that our current array with the same memristor device is still in use, we did not conduct the extensive endurance tests directly on the array which may damage the chip.

Figure R1. 10⁶ endurance cycles with different gate voltage in the 1T1M cell. (Adapted from Ref. [R4])

Reference:

R4. Merced-Grafals, E.J., et al., *Repeatable, accurate, and high speed multi-level programming of memristor 1T1R arrays for power efficient analog computing applications*. *Nanotechnology*, 2016. **27**(36): p. 365202.

7) In the benchmark in Table 1, it appears that the performance of the proposed memristor-based QPA is obtained by estimation on future implementations with 16nm technology node. Since a functional memristor macro is already fabricated in this work, please also show the benchmark results based on the experimentally measured data.

Response: We thank the reviewer for the suggestion. A brief discussion about the performance benchmarking results based on our current macro with 180 nm technology node has been added to *Supplementary Note S1* in revised SI for reference. However, it should be noted that our current hardware design is not fully optimized and is only used for proof-of-concept purposes. For an apples-to-apples comparison, it is better to refer to our estimation of future implementations with a 16 nm technology node.

In *Supplementary Note S1* in revised SI, we added:

“For our memristor-based QPA, the annealing time was set at 200 iterations, which results in a success probability of 16%. It takes 27 runs to reach a 99% success probability resulting in a time to solution of 5400 iterations. By using our current 180 nm technology node-based chip, the working frequency of which is 10Mhz (100 ns for each iteration) and the power dissipation of a single array in which is about 0.17 W. To run 5400 iterations, the estimated time to solution, energy to solution and solutions per second per Watt are 540 μs, 91.8 μJ and 1.09×10⁴, respectively. Notably, the current hardware design is not optimized and is used for proof-of-concept purposes. To ensure a fair comparison with other technologies, we estimated our performance based on a 16 nm technology node (see in Table S1). The calculated time to solution, energy to solution and solutions per second per Watt based on our new estimation are 10.8 μs, 24.8 nJ and 4.03×10⁷, respectively.”

8) The comparison of solution gap should be included in Fig. 2g.

Response: We thank the reviewer for the suggestion. Following the reviewer's suggestion, the comparison of the solution gap has been added as *Fig. S7* in the revised SI. Moreover, as the results of CIM and D-Wave are replotted from Ref. [R5], which does not include the comparison of the solution gap. Thus, we only compared the result obtained by QPA and SA here.

“Figure S7. Comparison of average solution gap (optimal maximum cut - obtained cut) of our QPA and SA.”

Reference:

R5. Hamerly, R., et al., *Experimental investigation of performance differences between coherent Ising machines and a quantum annealer*. Science Advances, 2019. 5(5): p. eaau0823.

9) It is recommended to add the units of computing error in Fig. 3d, and plot the curve of the measured MVM outputs with respect to the ideal MVM results directly.

Response: We thank the reviewer for the suggestion. Original *Fig. 3d* shows the normalized computing error, which does not have a unit, but we realize that it may cause confusion. To better reflect how accurately our hardware can perform MVM operations, the figure (*Fig. 3d*) has been modified in the revised main text to have a unit of μA , which has physical meanings. Moreover, following the reviewer's suggestion, the curve of measured MVM outputs with respect to the ideal MVM results has been plotted as *Fig. S8* in the revised SI.

In Fig. 3 in revised main text, we made the corresponding changes:

“d Distribution of computing error (the difference between the experimental result and the expected value) of the analog memristor crossbar, with the dashed line indicates a Gaussian distribution with a mean of $0.26 \mu\text{A}$ and a standard deviation of $17.19 \mu\text{A}$. The full output range is about -750 to $750 \mu\text{A}$. 1000 random input vectors with each entry randomly taken from either $+1$ or -1 were generated for the vector-matrix multiplication to get a distribution. The expected value is calculated by software with full floating-point precision. The direct plot of the experimental result with respect to the expected result can be found in Fig. S8.”

We added Fig. S8 to the revised SI:

“Figure S8. Experimental vector matrix multiplication result v.s. software calculated expected result. The gray dash line shows the line that experimental result perfectly matches the expected result.”

REVIEWERS' COMMENTS

Reviewer #1 (Remarks to the Author):

The authors have revised the manuscript satisfactorily - good with me.

Suhas Kumar

Reviewer #2 (Remarks to the Author):

The authors have fully addressed my comments and the paper can be accepted now for publication.

REVIEWERS' COMMENTS

Reviewer #1 (Remarks to the Author):

The authors have revised the manuscript satisfactorily - good with me.

Suhas Kumar

Response: Thank you for your time and efforts in reviewing our manuscript. We are pleased to hear that you find the revisions satisfactory. We appreciate your constructive feedback that enhanced the quality of our work.

Reviewer #2 (Remarks to the Author):

The authors have fully addressed my comments and the paper can be accepted now for publication.

Response: Thank you once again for your valuable input and for supporting our work.